# Piezoelectric Micromachined Microphone with High Acoustic Overload Point and with Electrically Controlled Sensitivity [note 1]

**DOI:** 10.3390/mi15070879

**Published:** 2024-07-03

**Authors:** Libor Rufer, Josué Esteves, Didace Ekeom, Skandar Basrour

**Affiliations:** 1ADT MEMS, 360 Rue Taillefer, F-38140 Rives, France; 2University Grenoble Alpes, CNRS, Grenoble INP, TIMA, 46 Av. Félix Viallet, F-38000 Grenoble, France; josue.esteves@univ-lyon1.fr (J.E.); skandar.basrour@univ-grenoble-alpes.fr (S.B.); 3Microsonics, 39 Rue Des Granges Galand, F-37550 Saint Avertin, France; didace.ekeom@microsonics.fr

**Keywords:** aeroacoustics, piezoelectric transducer, microphone, AOP, design, sensitivity control

## Abstract

Currently, the most advanced micromachined microphones on the market are based on a capacitive coupling principle. Capacitive micro-electromechanical-system-based (MEMS) microphones resemble their millimetric counterparts, both in function and in performance. The most advanced MEMS microphones reached a competitive level compared to commonly used measuring microphones in most of the key performance parameters except the acoustic overload point (AOP). In an effort to find a solution for the measurement of high-level acoustic fields, microphones with the piezoelectric coupling principle have been proposed. These novel microphones exploit the piezoelectric effect of a thin layer of aluminum nitride, which is incorporated in their diaphragm structure. In these microphones fabricated with micromachining technology, no fixed electrode is necessary, in contrast to capacitive microphones. This specificity significantly simplifies both the design and the fabrication and opens the door for the improvement of the acoustic overload point, as well as harsh environmental applications. Several variations of piezoelectric structures together with an idea leading to electrically controlled sensitivity of MEMS piezoelectric microphones are discussed in this paper.

## 1. Introduction

The field of micromachined microphones has seen significant advancements in recent years with the growing demand for miniaturization and high-performance sensing capabilities in various applications. For a long time, micro-electromechanical-system-based (MEMS) acoustic sensors have been the focus of academic and industrial research teams. The first developments of micromachined microphones were enabled by the progress in material science, fabrication technologies, miniaturization, and sensor techniques. Examples of these preliminary steps are the invention of the electret microphone [1] and progress in silicon-based static pressure sensors [2]. Further developments of micromachined microphones were achieved by many research teams and focused on the most common general approach using a diaphragm as an active microphone element converting the acoustic signal to the mechanical signal, and then converting the mechanical signal to the electric signal through known transduction principles. This effort resulted in the first microphones using piezoelectric [3], capacitive [4], and piezoresistive [5] couplings. Later, the FET (field effect transistor) microphone using a new principle, enabled only by silicon micromachining, was invented [6]. Finally, an optical microphone was invented, in which a diaphragm and a rigid structure form an optical waveguide with the geometry, and thus the transmission properties dependent on the diaphragm deflections modify the intensity and the phase of the transmitting optical signal [7].

From these early demonstrated micromachined devices, a capacitive microphone has been adopted as the dominant microphone type for further development for several reasons. The first reason was its much lower noise floor compared to piezoelectric or piezoresistive microphones. The other reason was the applicability of currently used industrial microtechnologies for its fabrication, with no requirement for additional structural layers or process steps. With strong industrial support, the micromachined capacitive microphone reached a commercial form after more than twenty years of incubation and became one of the most successful commercial MEMS products in the history of microsystem technology [8]. The continuing research of this kind of microphone resulted in key performance parameters such as sensitivity, signal to noise ratio, and distortion, meeting high requirements for microphones for mobile applications [9].

More recently, the availability of aluminum nitride (AlN) layers in industrial fabrication processes brought an increased interest in piezoelectric thin-plate-based micromachined devices. The following two kinds of acoustic devices using this new AlN-based technology were developed almost simultaneously: piezoelectric micromachined ultrasonic devices (PMUTs) and piezoelectric microphones.

PMUTs have been presented as counterparts of already mature capacitive micromachined ultrasonic devices (CMUTs). These kinds of thin-film-based micromachined ultrasonic transducers (MUTs) have been investigated as an alternative to conventional bulk, or thick-film, or piezocomposite ultrasonic transducers due to the advantages offered by microsystem technologies, namely small size, low power consumption, easy interconnexion, batch fabrication, and low cost. The working principle of CMUTs, like that of capacitive microphones, relies on the conversion of mechanical energy into electrical energy through an electrostatic field between two electrodes of a device. The energy conversion can take place in both ways, so the device can both transmit and receive acoustic signals. CMUTs have been demonstrated to work efficiently for both air and immersion applications and their main use is in medical imaging, but also in underwater imaging and nondestructive evaluation [10]. PMUTs exploit a 31-mode of a piezoelectric layer to generate or detect an acoustic signal. In cases of detection, an acoustic signal deforms the device diaphragm containing the piezoelectric layer and thus creates in-plain mechanical strain (1-direction), which results in an out-of-plane electric field (3-direction) obtained through the 31-transverse piezoelectric effect. The main applications of PMUTs are rangefinders [11], fingerprint sensors [12], and implantable micro-devices [13].

A piezoelectric microphone works, like a PMUT, in a 31-transverse mode. The main difference between the two devices consists of the useful frequency range definition. PMUTs are resonant devices and their useful frequency range is spread in a relatively narrow band around the resonant frequency. The useful frequency range of a microphone is located under its resonant frequency. The frequency range is limited at its high end by the resonant frequency and its low end is characterized by the leaks due to electrical and acoustic resistances. The piezoelectric microphone structure, thanks to the absence of a fixed electrode, offers a unique advantage due to its fabrication simplicity. The piezoelectric microphone can bring improvements in ruggedness and in moisture tolerance that are important in harsh environmental applications. Compared to the capacitive microphone, the piezoelectric structure enables higher diaphragm excursions, limited only by its nonlinear behavior, and thus higher acoustic overload point (AOP). This feature has attracted attention for industrial and aerospace applications with extremely high acoustic levels. One of the first piezoelectric microphones designed for aeroacoustic applications reached an AOP of 172 dB, which is substantially higher than in the currently available typical micromachined capacitive microphones [14].

The aim of this paper is to introduce modeling approaches and main design considerations related to MEMS piezoelectric microphones, and to compare and discuss simulation results for piezoelectric microphones with a circular diaphragm with a piezoelectric layer located in two specific diaphragm regions (one is close to the center and the other is close to the clamped edge). Based on the parametric optimization, the effects of several design parameters, such as the piezoelectric layer thickness and width, the sensing electrode localization, and the pressure equalization hole dimensions, on the microphone sensitivity and signal-to-noise ratio (SNR) are demonstrated in a typical case study. The optimization towards a high AOP is presented and the expected performance of this parameter is estimated.

The ability of a piezoelectric layer to serve not only as a sensor but also as an actuator opens the space to advanced designs of acoustic devices aimed at the control of various parameters such as resonant frequency, stiffness, or sensitivity. Various approaches using piezoelectric layers have been applied to acoustic and ultrasonic micromachined devices to achieve resonant frequency tuning [15,16] or sensitivity improvement [17]. A configuration with a piezoelectric layer with two distinguished sections, one used as a sensor and the other as an actuator, is also proposed in this paper. To the best of the authors’ knowledge, a similar solution demonstrating the diaphragm stiffening due to bias voltage, enabling microphone sensitivity and AOP tuning, has not been applied in the audio frequency range.

After this short overview of micromachined microphones, Section 2 presents the piezoelectric materials that are considered in the study, the microphone mechanical structure corresponding to the silicon on insulator (SoI) fabrication process, and the associated acoustic elements. Modeling approaches are described in Section 3, and optimization processes are explained in Section 4. Finally, Section 5 is devoted to the proposed microphone structure aimed at electrically controlled sensitivity.

## 2. Piezoelectric Microphone Structure

The microphone structure involves mechanical elements, neighboring acoustic elements and components introducing electromechanical coupling, enabling its main function—sensitivity to acoustic stimulation. The microphone’s performance is determined by the response of such a complete structure to an acoustic stimulus applied on its diaphragm.

The number and dimensions of each layer composing the microphone structure, as well as their material constants and associated stresses, depend heavily on the used fabrication technologies. The choice of the piezoelectric material was made in agreement with the technology applicable for its deposition and it also determined the appropriate materials for the bottom and top electrode layers.

### 2.1. Piezoelectric Layer

The function of a piezoelectric device requires a capacitor structure with a piezoelectric layer sandwiched between its top and bottom electrodes. If the main deformation of the piezoelectric layer obtained in diaphragm-based microphones is considered, the general tensor constitutive equation, coupling the electrical and mechanical domains, can be reduced to the following equations [18]:(1)$$S_{1}=s_{11}^{E}T_{1}+d_{31}E_{3},$$
(2)$$D_{3}=d_{31}T_{1}+\epsilon_{33}^{T}E_{3},$$
where *S*_1_ and *T*_1_ are the mechanical strain and stress in axis *1*, *E*_3_ and *D*_3_ are the electric field and the electric density displacement in axis *3*, *s*_11_*^E^* is the compliance constant at a constant electric field, *d*_31_ is the piezoelectric constant, and *ε*_33_*^T^* is the permittivity of the piezoelectric material at a constant mechanical stress. The components of Equations (1) and (2) correspond to the fact that the mechanical strain and stress are applied in the lateral dimension, which is perpendicular to the polarization, to the electric field and to the electric density displacement axis. Such a situation is described by the ‘*31*’ components of the piezoelectric matrix and by the corresponding so-called ‘*31*’ coupling mode of operation for piezoelectric materials.

In most of the applications using the bending of a piezoelectric thin film, the total elastic properties of a bending structure are often dominated by a substrate, which brings the main difference to the evaluation of the piezoelectric activity compared to bulk materials. The anisotropic interaction between the piezoelectric film and the substrate results in identical strains along in-plane directions (*S*_1_ and *S*_2_), and the stress perpendicular to the film surface is *T*_3_ = 0. Such a situation enables the derivation of effective piezoelectric coefficients and an example is shown below [19]:(3)$$e_{31,f}=\frac{d_{31}}{s_{11}^{E}+s_{12}^{E}}.$$

Compared to its intrinsic bulk value, the absolute value of the effective e-coefficient is always larger than *e*_31_. The effective piezoelectric coefficients can be conveniently used as an evaluation index for the piezoelectric characteristics of thin films. These coefficients can also be measured directly by unimorph cantilever-based methods [20]. Nevertheless, for numerical simulation-based modeling, all materials building the microphone structure must be described with complete matrices of intrinsic elastic, electric, and piezoelectric coefficients.

In this paper, an AlN layer is considered as a basic piezoelectric component of a microphone structure. In specific arrangements requiring an actuation function, the ferro-electric material lead zirconate titanate (Pb(Zr,Ti)O_3_), abbreviated as PZT, is also taken into account. The main matrix components describing piezoelectric materials involved in our study related to the ‘*31*’ coupling mode of operation are listed in Table 1.

The material parameters displayed in Table 1 depend heavily on physical parameters applied during the fabrication process and good knowledge of these parameters is critical for obtaining accurate simulation results. For the modeling purpose of this paper, focusing various simulation cases, Table 1 was drawn from the reference literature [14,21,22,23].

Various figures of merit focusing on various criteria have been adopted for piezoelectric transducers with bending elements. If the transmission of the acoustic signal is focused, the effective transverse piezoelectric constant can be used as a figure of merit for the transmission case.
(4)$$M_{T}=\left|e_{31,f}\right|.$$

Materials with a higher value of *M_T_* can produce larger sound pressure at the same driving voltage, or the required driving voltage becomes lower to obtain the same pressure level [24].

In a sense mode, when the membrane is deflected due to an impinging acoustic wave, a piezoelectric *g* constant is important [24]. A corresponding figure of merit for a sense mode is shown below.
(5)$$M_{S}=\left|\frac{e_{31,f}}{\varepsilon_{33}\varepsilon_{0}}\right|.$$

Another figure of merit representing the intrinsic signal-to-noise ratio of the material has been defined as follows [23]:(6)$$M_{N}=\left|\frac{e_{31,f}}{\sqrt{\varepsilon_{33}\varepsilon_{0}tan\delta }}\right|.$$

If both materials considered in this study are compared with the aid of the figures of merit (4) and (5), AlN could be effectively applied to sensors, whereas PZT is better suited for actuation purposes. The fast and simple comparison using the figures of merit was confirmed by using multi-criteria decision-making (MCDM) material selection techniques [25]. From the currently available piezoelectric materials, AlN clearly stands out as the best candidate for use as a microphone sensitive layer. For voltage detection, aluminum nitride leads in the quantitative parameters with its low dielectric constant, high resistivity, low loss tangent, and high SNR values. Moreover, AlN has good compatibility with complementary metal oxide semiconductor (CMOS) processing and good process quality control in manufacturing, which is important for device scaling and commercial applications. However, if current detection is preferred for sensors, or if force performance is required in actuating applications, the PZT appears as a clear leader among piezoelectric materials [21].

### 2.2. Mechanical Body

The purpose of the modeling and simulation work exposed in this paper is to present the effect of various design parameters on the microphone performance, and to propose a microphone structure aimed at electrically controlled sensitivity. For the sake of clarity, and to allow presenting main behavioral tendencies without secondary effects, a simplified basic wafer, shown schematically in Figure 1, was considered for microphone modeling in the first approach. Later, important effects due to additional layers required by a chosen fabrication process must have been included in the model.

The structure of Figure 1 consisting of the SoI wafer with deposited piezoelectric and metallic layers was chosen as a good alternative to the wafer fabrication process. The top silicon layer (device layer) can be chosen for the exact thickness, crystal orientation, and conductivity required by the application, and the buried oxide layer provides brilliant etch stop characteristics. The thickness of each wafer layer is freely adjustable and it can be, depending on a producer, in the range of 340 to 725 µm for the handle layer, 0.5 to 3 µm for the buried oxide layer, and 1 to 300 µm for the device layer. During the fabrication process of the microphone structure, the front side patterning is applied on the metallic and piezoelectric layers to form the microphone sensitive parts. These sensitive parts are formed by a sandwich composition in which the piezoelectric layer is placed between two electrodes. In this study, the top electrode is formed of a metallic layer (aluminum), and a silicon device layer serves as the bottom electrode. The microphone diaphragm can be constructed using backside etching of the silicon handle layer and buried silicon oxide layer.

High-quality piezoelectric films cannot be grown directly on silicon. Depending on the piezoelectric material and on its deposition process, inter-layers are necessary to provide an optimum nucleation rate or growth direction to prevent interdiffusion and oxidation reactions or to improve adhesion.

In the case when AlN is used as a piezoelectric layer, commonly used underlying materials include Pt, Ti, Al, and Mo. Platinum underlying layers have demonstrated their ability to grow high-quality AlN, due to their inertness to nitrogen. However, it is not used in most applications due to cost and patterning difficulties. Molybdenum is the most common material used as the underlying seed layer, which ensures the high quality of the piezoelectric layer and of the electrical contact [26].

The PZT films for most applications are grown on an electrode, which should neither oxidize nor become insulating. The most often reported materials include Pt, and metal oxides. Usually, the chemical barrier function is provided by two or more layers, including the electrode. PZT/Pt/Ti/SiO_2_/Si is the most widely applied sequence, in which titanium is needed as an adhesion layer [21,27].

The residual stress in structural layers appearing as one of the most common outcomes of the integration of distinctly different materials must be well controlled during the fabrication process. Even if strong consequences of residual stresses such as creep, deformation, fracture, or fatigue are avoided, they can still affect the elastic properties of the structure and have a strong influence on the final behavior of the device. Hence, the assessment and regulation of residual stress are one of the prime challenges to predicting the final performance of MEMS devices [28].

The residual stress in devices based on SoI-MEMS technology arises primarily from the residual stress in the SoI wafer itself and from the residual stress formed during the additional process to achieve the final device. Silicon direct bonding technology, used in the preparation of SoI wafers, involves annealing and thermal oxidation steps, inducing the residual stress generation within the wafer layers. The gradual release of the silicon handle layer and buried SiO_2_ layer disrupts the original stress balance mechanism within the SoI structure and leads to the development of tensile residual stress in the released silicon device layer. A mechanical theoretical model for the residual stress in SoI wafers was established and verified through experimental characterization and gives values of residual stress in the device layer in the range of 30 MPa [29].

The residual stress of additional processes is dependent on the materials deposited and on deposition conditions, and its final stress levels are known only in a relatively large interval of values. The thin film stress in polycrystalline AlN can range from compressive to tensile stress levels depending on the deposition technique and the parameters used. As an example, the residual stress in AlN thin films sputter-deposited in identical conditions on Si substrates was found to be compressive and its values were in the range of −300 (±50) MPa to −730 (±50) MPa. The difference in residual stresses can be attributed to the microstructure of the films and mismatch between in-plane atomic arrangements of the film and substrates with various orientations, including (111), (100) or (110) [30].

It is important to minimize the residual stress generated inside the device structure to minimize its effect on the performance, reliability, and yield. Simple compensation techniques to lower the overall stress are not sufficient, as AlN often exhibits a stress gradient along the thickness of the thin film. An example of a modified sputter process exploiting the influence of varying sputter pressure during deposition on the intrinsic stress component is presented in [31]. In the process, AlN thin films were synthesized with a DC magnetron sputter system at a temperature below 100 °C on p-type (100) silicon wafers. The back pressure of the pure nitrogen atmosphere in the sputter chamber was applied in two specific phases to reliably fine tune the resulting stress to −170 MPa, while keeping a high piezoelectric coefficient.

In the previous paragraphs, it was demonstrated that detailed knowledge of all components of the MEMS structure including the residual stress is highly important for accurate device simulation and design phases. It was also shown that with stress engineering, structures with piezoelectric films with low residual stress can be attained, but their study and elaboration are beyond the scope of this paper. For the simulations, residual stresses were not considered, and the simplified structure of Figure 1 was used, which is well in line with the purpose of the paper to present various microphone configurations and their design parameters. Table 2 summarizes the main material constants values of passive layers that were used in the work.

It can be noted that although isotropic materials are typically described with two engineering constants as Young’s modulus and Poisson’s ratio, here the compliance matrix elements are presented to keep uniformity with Table 1.

In this study, the simulation results of the three axisymmetric microphone structures shown in Figure 2 are compared. The colors of the structure layers used in Figure 2 are identical to those described by the legend of Figure 1. Firstly, a microphone (type A, shown in Figure 2a) with a circular diaphragm, as described in [14], with the sensing electrode in the proximity of the clamped diaphragm edge is studied. In another microphone structure (type B, shown in Figure 2b), the diaphragm has its sensing electrode located around its center. Finally, the study is completed with a microphone structure (type C, shown in Figure 2c), exploiting both electrodes, including a peripheral and a central electrode [32]. This configuration can be used in two ways. Firstly, both electrodes are used as sensing electrodes and the microphone output is obtained as a difference in both electrode signals. Another method described in the paper consists of using one of the electrodes as a sensor and the other one as an actuator helping to electrically control the sensitivity of the overall structure.

### 2.3. Acoustic Environment

Acoustic environment is a crucial part of the acoustic sensor. It is important not only as the propagation medium, ensuring the interaction between a sound source and the outer side of the diaphragm, representing the microphone input, but also for building elements placed inside the microphone body, communicating with the inner side of the diaphragm. Acoustic impedances of these elements must be taken into account together with mechanical impedances of structural parts of the microphone. Here, the main acoustic elements related to the microphone model are introduced, and simplified expressions corresponding to our design in terms of dimensions and the frequency range are presented. More detailed theory of this field can be found elsewhere [33].

The acoustic impedance Z_a_ is the ratio of sound pressure *p* (in Pa) to volume flow rate *q* (or volume velocity in m^3^/s). Using the electroacoustic analogy, basic acoustic elements can be defined through simple structures presented by cavities and ducts.

If acoustic pressure *p* is applied in a small cavity with the volume *V*, with dimensions much smaller than the wavelength *λ*, the fluid (air) in the confined volume acts like a spring. In an analogy with mechanical compliance, a compact enclosed cavity is called an acoustic compliance with the following value [33]:(7)$$C_{a}=\frac{V}{\rho_{0}c_{0}^{2}},$$
where *ρ*_0_ is the fluid density and *c*_0_ is the speed of the sound. A cavity placed underneath the diaphragm is an important part of a piezoelectric microphone. Its compliance can be fixed in a large range of values, and thus can tune the resulting resonance frequency of the microphone.

If a duct or pipe with a rectangular cross sectional area A and length *L* ≪ *λ* are considered, the fluid in the duct vibrates due to an acoustic pressure difference *p* applied across it. Such a component presents a complex acoustic impedance composed of the acoustic inertance (mass) with of the following value [33]:(8)$$M_{a}=\frac{\rho_{0}L}{A},$$
and the acoustic resistance with the following value [33]:(9)$$R_{a}=\frac{12\mu L}{bh^{3}},$$
where *μ* is the viscosity of the fluid, and *b* ≫ *h* are the width and height of the duct. These elements are used to model the flow through the pressure equalization vent channel of the microphone.

The acoustic pressure *p* at the surface of the diaphragm, which can be approximated by a rigid piston, is the pressure due to the impedance *Z_rad_* of the radiation field. The radiation impedance in the low-frequency approximation (*ka* ≪ 1, where *k* = *ω*/*c*_0_, and *a* is the piston radius) can be simplified as a series combination of the radiation mass *M_ad_*, *r_ad_* and the radiation resistance *R_ad,_**_rad_* [33].
(10)$$M_{ad,rad}=\frac{8\rho_{0}}{3\pi^{2}a},$$
(11)$$R_{ad,rad}=\frac{\rho_{0}\omega^{2}}{2\pi c_{0}}.$$

In diaphragm-based microphones, thin film deformations induced by acoustic pressure play an important role in the device. It was verified that the corresponding stresses present in the microphone structure are significantly lower than the materials’ tensile strengths.

## 3. Microphone Modeling

### 3.1. Finite-Elements Model

The modeling presented in the paper relies mainly on the following two approaches: finite-element modeling (FEM) and lumped-element modeling (LEM). The finite-element model was developed in the Ansys Workbench ver. 2022 R1. The proper definition of the boundary and symmetry conditions facilitates modelling of only a portion of the actual structure, which reduces the analysis run time and memory requirements with no losses in accuracy. Most of the simulations were carried out on one quarter of the structure, and in some cases, one eighth of the structure was used to speed up the calculation time. Structural layers (Si, SiO_2_ and Al) are meshed with SOLID 186 elements; the piezoelectric layer (AlN) is meshed with SOLID 226 elements, allowing us to define piezoelectric properties. Meshing size varies with the dimensions of the structure during the optimization process. Nevertheless, working with the linear lumped-element model towards the complete microphone evaluation was preferred for speed and availability purposes. FEM was thus an important tool to define the lumped elements with high accuracy. It is briefly shown in Section 4.5 how FEM was used to evaluate the nonlinear behavior of the microphone diaphragm.

FEM also enables us to verify the stress situation in the structure and to predict the deformation and stress fields when intrinsic stress is present. It was shown in [34] that several restrictions exist in commercially available tools to combine a piezoelectric analysis with intrinsic stress in a harmonic response analysis. The arising difficulties could be overcome by employing a coupled thermo-electromechanical simulation in order to obtain consistent static and harmonic response results. A 3D FEM including intrinsic stress effects must be considered in advanced design, but this is not the focus in this paper.

### 3.2. Lumped-Element Model

Lumped-element modeling is a powerful and reliable method for predicting the multiphysics behavior of electroacoustic transducers. With this method, each element of the transducer is transformed to a circuit model thanks to the mechanical (mass–damping–stiffness) and electrical (inductor–resistor–capacitor) equivalence. Using this method requires characteristic lengths of the system smaller than the wavelength of the associated physical phenomena, which is satisfied in the audio-frequency range. Here, the LEM shown in Figure 3 already presented in [14] is used. This model allows evaluation of the microphone performance including its frequency range, sensitivity, noise, SNR, and minimum detectable pressure (MDP) values.

The model represents the mechanical elements transformed to its acoustic side (resistance, mass and compliance of the diaphragm, *R_ad_*, *M_ad_*, *C_ad_*), acoustic elements (diaphragm radiation mass and resistance, *M_ad_*_,*rad*_, *R_ad_*_,*rad*_, mass and compliance of the back cavity, *M_ac_*, *R_ac_*, and the pressure equalization vent resistance, *R_av_*), and the electrical elements (sensing and parasitic capacitance, *C_eb_*, *C_e_*_0_, resistance of the piezoelectric layer and of the connection lines, *R_ep_*, *R_es_*). The acoustic pressure at the microphone input *p* is transformed to the output electrical voltage *v*_0_ with the transducer factor *Φ*_0_.

The localized constants *M_ad_*, *C_ad_*, and *C_eb_* are extracted from the finite element calculation, without making any assumptions about the geometry or the shape of the eigenmode. In the vicinity of the resonance, the effective mass *M_ad_* of the microphone (in the *z* direction, normal to the membrane) is given by the following equation [35]:(12)$$M_{ad}=\frac{U^{T}\left[M\right]U_{z}}{U^{T}\left[M\right]U},$$
where [*M*] designates the finite element global mass matrix; *U* is the eigenvector associated with the natural frequency *f_r_* of the microphone. The displacement vector *U_z_* is such that all its components in the *z* direction are equal to *1* and its other components are equal to *0* in the vicinity of the resonance frequency. The effective mass *M_ad_* in the *z* direction is linked to the effective stiffness *K_ad_* and to the resonance frequency *f_r_* by the following relationship [35]:(13)$$\frac{K_{ad}}{M_{ad}}=\frac{U^{T}\left[K\right]U}{U^{T}\left[M\right]U}={\left(2\pi f_{r}\right)}^{2},$$
where [*K*] is the global finite element stiffness matrix. It follows that *K_ad_* is given by the following equation [35]:(14)$$K_{ad}={\left(2\pi f_{r}\right)}^{2}M_{ad}=\frac{1}{C_{ad}}.$$

The blocked capacitance of the transducer, *C_eb_*, is obtained from the following equation [35]:(15)$$\frac{1}{2}\Phi^{T}\left[K_{\Phi \Phi }\right]\Phi =\frac{1}{2}C_{eb}V^{2},$$
where *K_ΦΦ_* is the overall dielectric matrix, *V* is an arbitrary voltage applied to the hot electrode and *Φ* is the overall voltage vector, such that the voltage is equal to *V* for the nodes located on the electrode and *0* for the other nodes. The resulting relation for *C_eb_* is as follows [35]:(16)$$C_{eb}=\Phi^{T}\left[K_{\Phi \Phi }\right]\Phi .$$

The remaining lumped elements, including *M_ac_*, *C_ac_*, *R_av_*, *M_ad_*_,*rad*_, and *R_ad_*_,*rad*_, were obtained analytically from the expressions presented in Section 2.3. This work is focused on the presentation of the main structures applicable to piezoelectric microphones and on the comparison of their performance parameters. The parameters, such as the resonant frequency, the sensitivity, SNR, and AOP, used for the comparison are well known in the field of acoustic sensors, and thus they are not presented here in detail. More information on these parameters can be found elsewhere [14,36].

## 4. Parametric Optimization of the Microphone Structure

In piezoelectric microphones, thickness and lateral dimensions of all diaphragm layers, including piezoelectric and electrode materials, are key parameters that need to be optimized in the design loop. Typically, there is not a unique optimal solution satisfying the microphone specifications, and widely used optimization algorithms often lead to trivial solutions. For this reason, parametric optimization already used in [37] was applied. The optimization process is performed in two computing environments (Ansys 2022 R1 and Matlab R2023a) and involves two selection levels. This approach helps to limit the computational requirements by eliminating simulation cases not satisfying the condition of the first selection level, and thus to limit the number of numerical simulations entering the second selection level. The conditions corresponding to both selection levels depend on the focused parameters of the specifications.

By default, the cases discussed in this section are based on the structure presented in Figure 1 and on the variant (a) in Figure 2 with the use of the AlN piezoelectric layer. Any deviation from this plan will be announced when necessary. The results are presented with the aim of giving the main design tendency and for this reason, not all dimensions and details are strictly listed.

### 4.1. Optimization of the Piezoelectric Layer Thickness

In order to show the effect of the piezoelectric layer thickness on the microphone sensitivity and signal-to-noise ratio, the optimization process of the whole microphone structure with the optimal design of a microphone matching the audio frequency bandwidth is run first. To carry out this, the limits for the structure dimensions are set and static and modal finite-element analyses (FEA) are executed. The obtained results are filtered through the No. 1 condition of the optimization process. This condition is satisfied only by the solutions with the frequency of the first mode in a predefined range, which are related to the maximal frequency of the bandwidth. A number of suitable designs, depending on the range size defined for the first condition, passes back to Ansys for the lumped element extraction based on the static simulation, and then to Matlab for the evaluation of the microphone characteristics. For the final step, the maximal value of the SNR is selected as the No. 2 condition. The frequency response of the optimized geometry is then compared with electromechanical harmonic analysis from Ansys [37]. The thickness of the piezoelectric layer can be fine-tuned in an additional step, which gives the results shown in Figure 4. In the example shown in Figure 4, the diaphragm diameter was 880 µm and the thickness of the piezoelectric layer was considered in the range from 0.1 µm to 2 µm.

If both graphs of Figure 4 are compared, it is evident that, in this case study, the maximal microphone sensitivity is achieved when the thickness of the piezoelectric layer is close to 1 µm, whereas the maximal SNR is reached when the piezoelectric layer thickness is 0.4 µm. This difference is due to the dependence of the noise density on the total electrical impedance of the piezoelectric layer and on the thickness at which it reaches its minimal value.

### 4.2. Optimization of the Piezoelectric Layer Width

The optimization of the piezoelectric layer width is performed on two similar structures illustrated in Figure 5. The only difference between these two cases is the fact that the structure in Figure 5a has the piezoelectric layer deposited on the whole diaphragm surface, while in Figure 5b, the piezoelectric layer is patterned in the same way as the top metallic layer. This difference may become important in view of the fabrication process, as the ‘full piezo’ layer from Figure 5a will require fewer etching steps than the ‘ring piezo’ layer from Figure 5b.

The optimal width of the piezoelectric layer can be found based on a similar process as in the previous paragraph. The primary structure, optimized for the audio frequency bandwidth, is subjected to an additional optimization step in which the width of the piezoelectric layer is fine-tuned. The main layers of this primary structure have a diameter of 880 µm and the thickness of the silicon layer is 3 µm and that of the piezoelectric AlN layer is 1 µm. The obtained results shown in Figure 6 document, again, that the sensitivity and SNR culminate at different values of the electrode width. Nevertheless, it is possible to find a compromise width, for which both parameters maintain reasonably high values. Moreover, the structure with the piezoelectric layer covering the whole diaphragm surface, which is technologically more suitable, shows substantially better sensitivity and SNR.

### 4.3. Effect of the Sensing Electrode Localization

According to the plate bending theory, if an originally stress-free thin circular plate rigidly clamped on its boundary is uniformly charged, two zones with opposite stresses can be distinguished on its surface. Both zones are delimited with a circle of radius *r* equal to the following [38]:(17)$$r=a\sqrt{\frac{1+\nu }{3+\nu }},$$
where *a* is the plate radius and *ν* is the Poison’s ratio of the plate material. Expression (17), predicting a change in curvature and thus a change in stress polarity located at approximately 60% of the plate radius for currently used materials, can serve only as a rough estimation of electrode placement. Advanced mechanical analysis of the multi-layer circular composite plate including a piezoelectric layer and initial stresses for the range of parameters used in the microphone design was carried out in [39].

In this case study, the same structure dimensions as in Section 4.2 were applied. In Figure 7, the sensitivity and SNR of structures shown in Figure 2a,b are compared. Both configurations clearly demonstrate a sensitivity decrease for dimensions over the limit given by the Expression (17). Even if the sensitivity for the structure in Figure 2b is substantially more important than for the case of the structure in Figure 2a, both structures can reach a similar performance in terms of the SNR.

If the zones defined by Expression (17) are respected, the annular ring electrode and the circular central electrode are located on diaphragm sections exposed to an opposite polarity of stress induced by the acoustic pressure. This fact can be exploited by appropriately connecting both electrodes in series or parallel circuit configurations in order to optimize raw voltage versus capacitance trade-offs [32].

A similar consideration of induced charge distribution of opposite signs in the central and surrounding region of a piezoelectric ultrasonic transducer was exploited in [17]. The respective two charge signals can be summed up effectively by complementary connection of the capacitors corresponding to each region. Moreover, statically deflected diaphragms cause a larger lateral strain in the piezoelectric layer compared to flat diaphragms. Sensors combining these two effects have been designed and they have shown sensitivity over five times higher than that of a conventional sensor.

### 4.4. Effect of the Pressure Equalization Hole

An important part of the microphone is a venting system designed to equalize the static pressure on both sides of the diaphragm. The venting system is typically realized by connecting the inside cavity of the microphone to the exterior space by a capillary. As the capillary, representing the acoustic resistance with acoustic mass, together with the inside cavity, representing the acoustic compliance, create a high-pass filter, all dimensions must be carefully tuned in a way that only useless frequencies are cut off, and the required frequency behaviour does not deteriorate. Figure 8 compares the effect of the vent hole diameter on the frequency response of the microphone.

For the purpose of the example shown here, the standard structure from the previous paragraphs was used. In the simulation process, a minimal capillary length, corresponding to the total thickness of the Si device layer (3 µm), and AlN layer (1 µm), was considered and its diameter size from 1 µm to 10 µm was taken into account. It is worth noting that the curve in Figure 8 corresponding to the smallest capillary diameter is masked by the effect of the electrical resistance *R_ep_* of the piezoelectric layer, which results in the change in the frequency response slope in the low frequency range.

### 4.5. Optimization Towards the Acoustic Overload Point

If a microphone withstanding a high acoustic pressure is designed, it is necessary to set the limits for the structure dimensions and to determine, through the nonlinear static analysis, the maximal diaphragm displacements for a given range of the input static pressures. Based on the deviation between the obtained value of the nonlinear displacement and the corresponding linear displacement, the acoustic overload point is obtained [14,37]. Figure 9 shows that there is an important difference between the linear and the nonlinear displacement responses of the microphone structure. To evaluate the maximal displacement, the optimization condition No. 1 corresponding to the difference between the linear and the nonlinear displacement equal to 3% was set. Each structure fulfilling the No. 1 condition for a predefined range of acoustic pressures passes towards selection No. 2. In this second step, the maximal value of the SNR is used as the No. 2 condition. It can be expected that the microphone with the structure described in the paper, with a diaphragm diameter of 880 µm, can reach the AOP of 163 dB, sensitivity of 212 µV/Pa, and the resonant frequency of 88 kHz.

## 5. Microphone with Electrically Controlled Sensitivity

In this section, the idea of microphone sensitivity control through a DC bias voltage is outlined. To this aim, the structure shown in Figure 2c, with one electrode at the edge and the other at the centre, is exploited. A similar configuration is typically used in PMUTs, where one electrode is activated with the electrical signal to be emitted and the other one, associated with the sensing layer, provides the electrical signal corresponding to a received stimulus. Unlike PMUTs, for microphone sensitivity control, a DC bias voltage is applied to one of the electrodes to induce stress through the associated piezoelectric layer to control the diaphragm overall stiffness.

### 5.1. Basic Approach

As discussed in Section 4.3, each electrode is located in a clearly distinguished zone of the diaphragm, separated by the radius given by Expression (17). Two SoI-based structures were considered, one with the AlN, and the other with the PZT piezoelectric layers, optimized for a wide-frequency band, with a resonant frequency in the vicinity of 60 kHz. For a better demonstration of strains variations in the system, the residual stress was not considered in the models of both structures. Nevertheless, as it was already stated earlier, the components of the residual stress must be involved in future models to confirm the hypotheses described in this section.

In the first simulation steps, FEM static analyses were carried out in order to compare the diaphragms’ deformation profile due to the injection of a bias voltage to the external and the central electrodes, respectively. Figure 10 shows the displacement profiles of the diaphragms with AlN and PZT as a function of the bias voltage. Only half of the symmetrical diaphragm is shown, with the central point located in *0*. Due to the superior performance of the structure with the PZT layer, confirmed with the deformation profile, only this structure was considered for the next simulation step.

### 5.2. Simulation Results

The initial FEM static analysis confirmed that the bias voltage of ±10 V applied on a PZT layer results in a clear change in the diaphragm profile. Such a modification of the deformation curve gives evidence of a modified stress situation in the structure, which can lead to the resonant frequency shift and to the variation in the microphone sensitivity. Both these effects on microphone performance can be demonstrated through harmonic analyses simulated with various bias voltages. The resulting frequency characteristics, shown in Figure 11, allow us to expect that the sensitivity of this microphone structure can be modified in the range reaching 10 dB with the bias voltage up to 10 V. The results are focused on the sensitivity levels, and the effect of acoustic elements is not considered at this stage. The decay on the low-frequency side of the response is due to the dielectric losses in the piezoelectric layer, as was already mentioned in Section 4.4.

The electrically controlled sensitivity described above can be further enhanced by the initial buckling of the diaphragm. The relationship between static deflection of the diaphragm and the acoustic sensitivity was demonstrated by other teams. Diaphragm static deflection due to compressive residual stress can be randomly oriented upward or downward due to the bending moment of the diaphragm at the releasing step. The lateral strain of a diaphragm is caused by bending as well as by expanding due to the large static deflection.

These two kinds of strains on layers above the neutral plane are summed up in the upward-deflected diaphragm but cancel each other in the downward-deflected diaphragm. Supposing that the piezoelectric layer is located on top of the diaphragm structure, above the neutral plane, it is exposed to higher stress in the upward-deflected diaphragm than in the downward-deflected one. Detailed analysis of the multilayered diaphragm buckling due to its sensitivity and the methods for deflection control using the buckling process were proposed in [40].

## 6. Conclusions

In this paper, the modeling and optimization approaches applied to the structure of piezoelectric MEMS microphones are presented. The basic aspects to be considered in the micromachined microphone design are discussed on the basis of a typical case study. A realistic estimation of the microphone performance, obtained following the optimization towards the acoustic overload point, is in agreement with the currently published specifications for microphones required for aeroacoustic measurements in application fields where commercialized MEMS microphones are inconvenient. Piezoelectric MEMS microphones can thus satisfy requirements for applications in aircraft design and rocket launching vehicles, as well as for new applications including the detection of gunshots or screams for military and urban security systems, requiring a high sound pressure level detection and harsh environment resistance [41,42,43].

Finally, a new approach enabling the feasibility of microphone sensitivity control is introduced. This approach is based on the configuration with one circular electrode located in the diaphragm center and one annular electrode placed close to the diaphragm clamped edge. One of the electrodes serves as a sensor providing the microphone output signal, while the other electrode is used as an actuator that can, with a DC bias voltage, mechanically pre-stress the diaphragm and thus modify the microphone sensitivity. Such an electrically controlled sensitivity has strong potential for applications dealing with high acoustic loads that are already enumerated.

## Figures and Tables

**Figure 1 micromachines-15-00879-f001:**
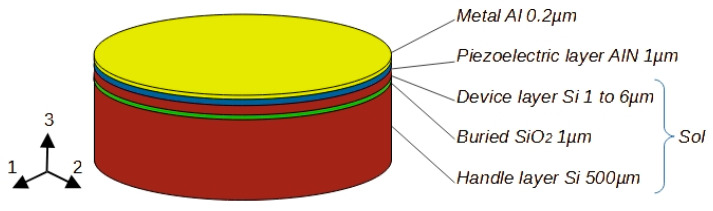
Basic SoI wafer with additional layers and their dimensions used for piezoelectric microphone modeling.

**Figure 2 micromachines-15-00879-f002:**
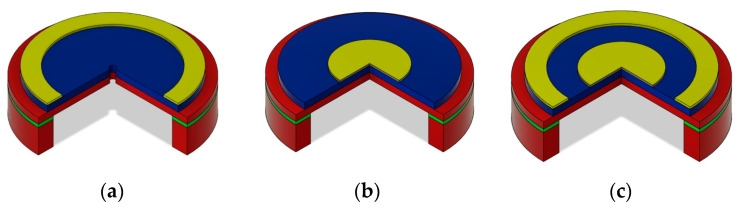
Mechanical structures of piezoelectric microphones used for the study schematically showing the electrodes (**a**) at the edge; (**b**) at the center; (**c**) both at the edge and at the center.

**Figure 3 micromachines-15-00879-f003:**
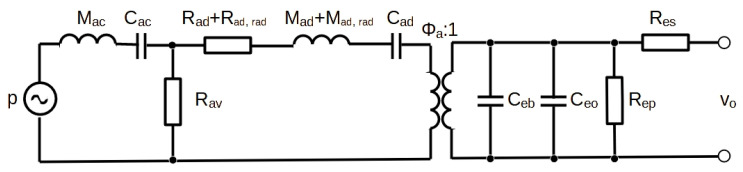
Lumped-element model of the piezoelectric microphone.

**Figure 4 micromachines-15-00879-f004:**
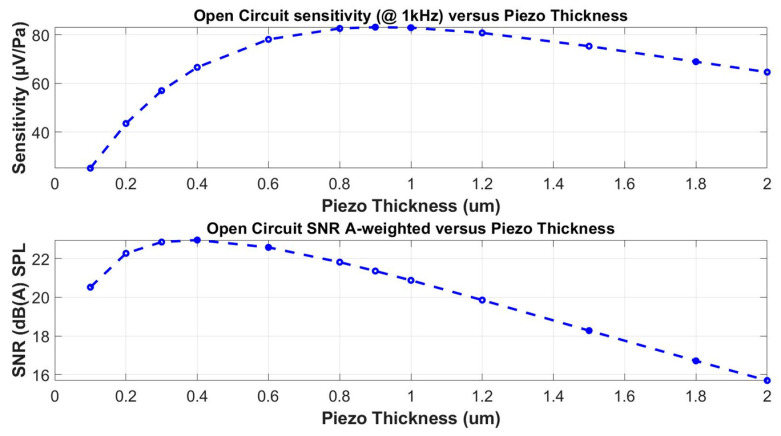
Effect of the piezoelectric layer thickness on microphone sensitivity and SNR.

**Figure 5 micromachines-15-00879-f005:**
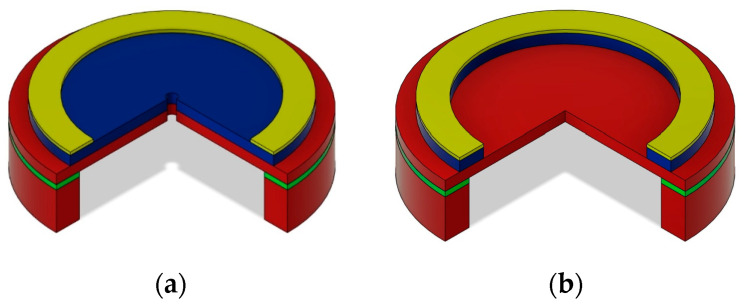
Microphone structure with the electrodes at the edge, with the piezoelectric layer deposited (**a**) on the whole diaphragm surface (full piezo); (**b**) under the metallic electrode (ring piezo).

**Figure 6 micromachines-15-00879-f006:**
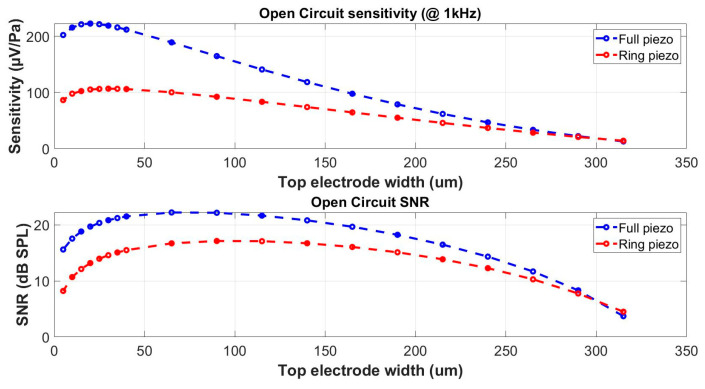
Effect of the electrode width on the microphone sensitivity and SNR with the piezo-electric layer fully covering the diaphragm (full piezo) or localized only under the electrode (ring piezo).

**Figure 7 micromachines-15-00879-f007:**
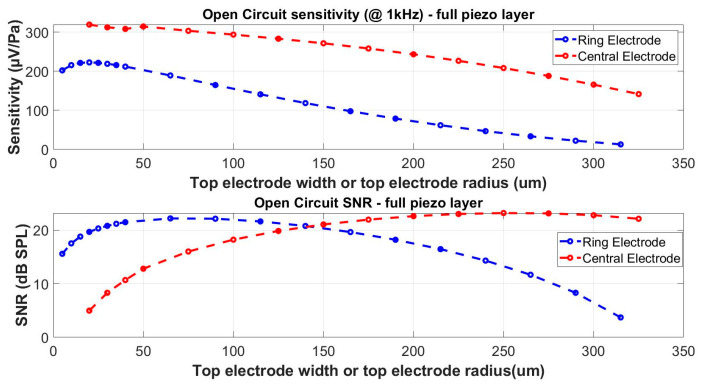
Comparison of piezoelectric microphones with metallic electrodes at the edge (ring electrode), and at the center (central electrode).

**Figure 8 micromachines-15-00879-f008:**
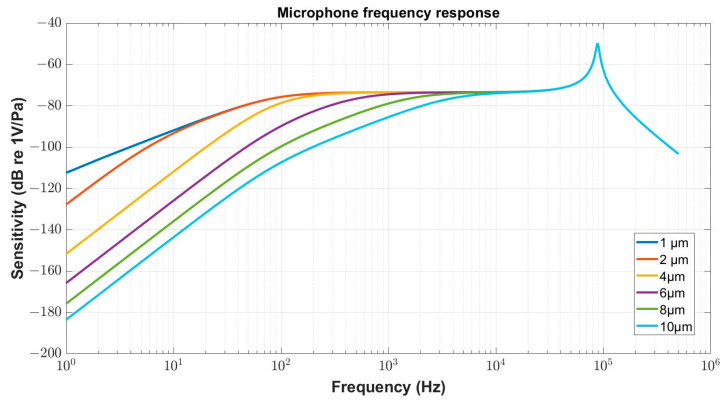
Effect of vent hole diameter on the microphone frequency response.

**Figure 9 micromachines-15-00879-f009:**
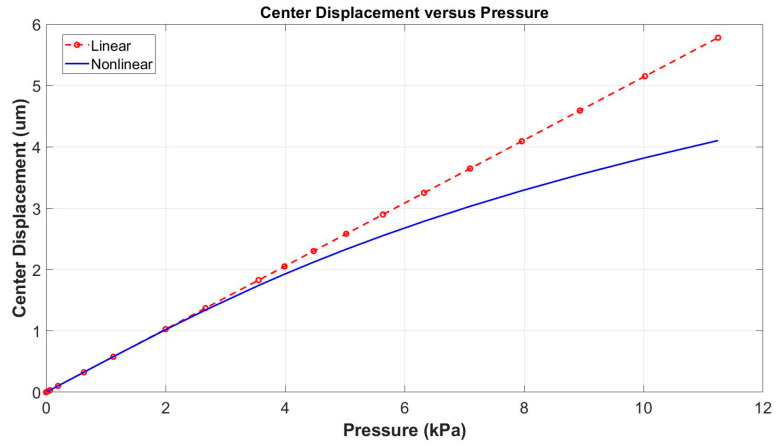
Maximal diaphragm displacement for the microphone structure obtained in linear and nonlinear static FEA.

**Figure 10 micromachines-15-00879-f010:**
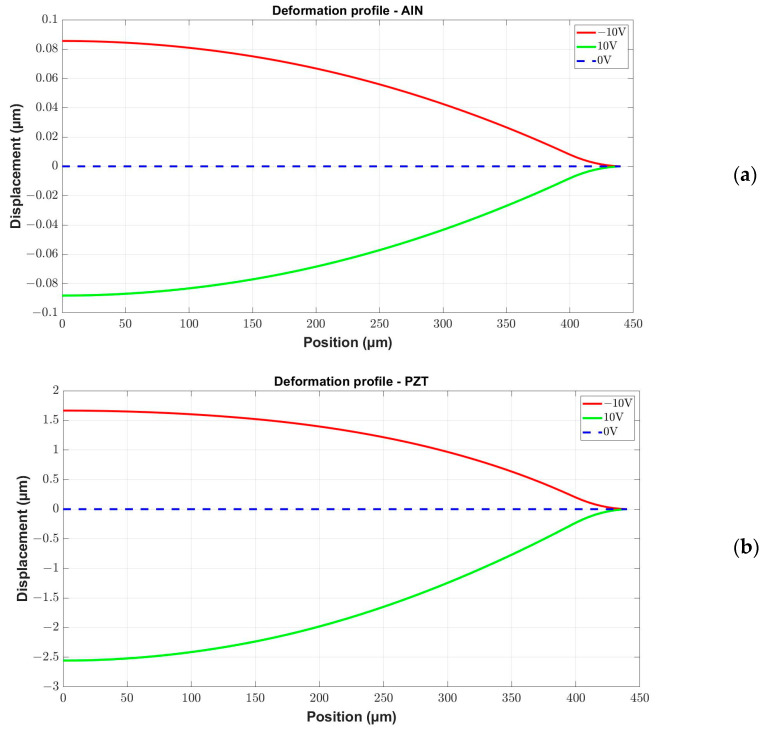
Displacement profile as a function of the static voltage applied on the ring electrode of the microphone diaphragm with the piezoelectric layer formed by (**a**) AlN; (**b**) PZT.

**Figure 11 micromachines-15-00879-f011:**
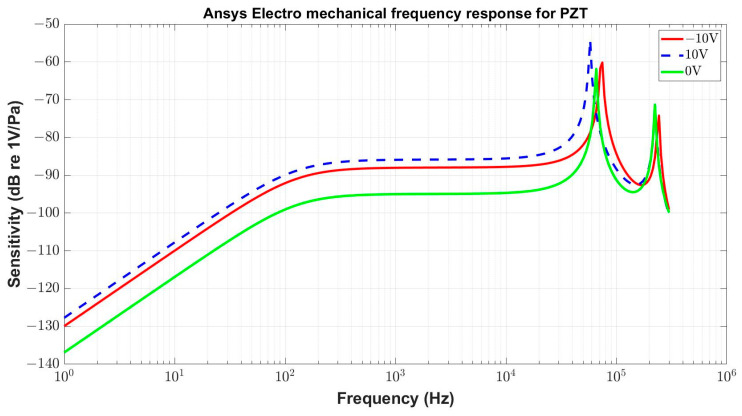
Frequency response of the microphone with the piezoelectric layer formed by PZT as a function of the static voltage applied on the ring electrode.

**Table 1 micromachines-15-00879-t001:** Main physical properties of piezoelectric materials used in the simulations.

Property	AlN	PZT
Compliance $s_{11}^{E}$ [TPa^−1^]	3.53	12.8
Compliance $s_{12}^{E}$ [TPa^−1^]	−1.01	−3.7
Compliance $s_{13}^{E}$ [TPa^−1^]	−0.77	−5.8
Material density [kg/m^3^]	3260	7700
Piezoelectric constant *d*_31_ [pm/V]	−2.65	−118
Piezoelectric constant *e*_31_ [C/m^2^]	−0.58	−4.1
Relative permittivity $\epsilon_{33}^{T}$ [-]	9.5	1160
Dielectric loss angle (tanδ) [%]	0.3	2
Figure of merit M_N_ [10^5^ Pa^1/2^]	20.9	9

**Table 2 micromachines-15-00879-t002:** Physical properties of passive mechanical layers used in the simulations.

Property	Si	SiO_2_	Al
Compliance $s_{11}^{E}$ [TPa^−1^]	7.67	13.7	14.3
Compliance $s_{12}^{E}$ [TPa^−1^]	−2.13	−2.33	−5
Compliance $s_{13}^{E}$ [TPa^−1^]	−2.13	−2.33	−5
Material density [kg/m^3^]	2330	2200	2700

## Data Availability

The original contributions presented in the study are included in the article, further inquiries can be directed to the corresponding author.

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
