# Peer review of "Piezoelectric Micromachined Microphone with High Acoustic Overload Point and with Electrically Controlled Sensitivityâ€"

_micromachines, 2024, doi:10.3390/mi15070879_

Round 1
Reviewer 1 Report
Comments and Suggestions for Authors
This paper presents a novel design of a piezoelectric microelectromechanical systems(MEMS) microphone with a high acoustic overload point (AOP) and electrically adjustable sensitivity. The proposed design shows potential for practical applications in high sound pressure environments. However, before making a final recommendation, there are several issues that need to be addressed by the authors.
1.The paper includes extensive simulation optimizations of device performance but lacks the remaining dimensions of the designed device when optimizing individual parameters. As an example, it is suggested to provide these dimensions, such as the piezoelectric layer diameter when optimizing its thickness.
2. The paper should include a performance comparison with existing similar types of piezoelectric MEMS microphones to more clearly highlight the advantages of the proposed device.
3.There are some minor issues in the paper that need to be improved.For example, the line graph and annotations in Figure 11 are very blurry. To ensure readers can clearly understand the information, the layout and clarity of the image should be enhanced.
Author Response
We appreciate the helpful comments and suggestions given by reviewers.
The manuscript has been revised accordingly, with all changes highlighted in yellow in the manuscript file.

Reviewer 2 Report
Comments and Suggestions for Authors
In this manuscript, the authors compare the simulation results for piezoelectric microphones with circular diaphragm. It is interesting and I recommend to accept after the revision.
1. The novelty of this paper needs more recent and comparative studies to support the claim. The references are very outdated and do not fully account for recent advancements in piezoelectric microphones.
2. This paper discussed the importance of residual stress but did not consider it in simulations for simplicity. This exclusion of residual stress can substantially impact the accuracy of the modeling, as it influences both the mechanical and piezoelectric performance of MEMS devices, as discussed by other researchers (Journal of Materials Science: Materials in Electronics 32.6 (2021): 6705-6741). Therefore, the reliability of these results is questionable without accounting for the effects of residual stress. Incorporating residual stress into simulations would likely provide a more accurate representation of the device performance.
3. Page No. 1, Line 23: Keywords: The authors should give at least five.
4. Remove the commo (,) from all the Equations. All the equations are derived? If not, the authors should cite the suitable references
5. Page No. 7, Line 291: “…b) at the center, c) both at the…” should change to “…b) at the center, and c) both at the…”
6. Figs. 3, 10 and 11 is not clear.
7. Line 251: “…(±50) MPa, to -730…” should change to “…(±50) MPa to -730…”
8. Line 253: “…(100), or (110)” should change to “…(100) or (110)”
9. Line 372: “… Cad and Ceb…” should change to “… Cad, and Ceb…”
10. The authors should cite suitable references inside the Table 1 and compare the recently reported papers.
11. Page No. 16, Line 548: “… by AIN a) and PZT b).” should change to “… by a) AIN and b) PZT.”
12. Page No. 16, Line 550: “… voltage of +/- 10 V…” should change to “… voltage of ± 10 V…”
13. The entire manuscript should be checked by a native English speaker.
Comments on the Quality of English LanguageAuthor Response
We appreciate the helpful comments and suggestions given by reviewers.
The manuscript has been revised accordingly, with all changes highlighted in yellow in the manuscript file.
